# Siberian Wildrye (*Elymus sibiricus* L.) Abscisic Acid-Insensitive 5 Gene Is Involved in Abscisic Acid-Dependent Salt Response

**DOI:** 10.3390/plants10071351

**Published:** 2021-07-02

**Authors:** Ying De, Fengling Shi, Fengqin Gao, Huaibin Mu, Weihong Yan

**Affiliations:** 1College of Grassland, Resources and Environment, Inner Mongolia Agricultural University, Hohhot 010011, China; deying@caas.cn; 2Chinese Academy of Agricultural Sciences, Grassland Research Institute, Hohhot 010010, China; gaofq1211@126.com (F.G.); sweethuai@yahoo.com.cn (H.M.); yanweihong7037@126.com (W.Y.)

**Keywords:** abscisic acid, Siberian wildrye, salt stress, *EsABI*5 gene, expression mechanism, functional identification

## Abstract

Siberian wildrye (*Elymus sibiricus* L.) is a salt-tolerant, high-quality forage grass that plays an important role in forage production and ecological restoration. Abscisic acid (ABA)-insensitive 5 (*ABI5*) is essential for the normal functioning of the ABA signal pathway. However, the role of *ABI5* from Siberian wildrye under salt stress remains unclear. Here, we evaluated the role of *Elymus sibiricus* L. abscisic acid-insensitive 5 (*EsABI5*) in the ABA-dependent regulation of the response of Siberian wildrye to salt stress. The open reading frame length of *EsABI5* isolated from Siberian wildrye was 1170 bp, and it encoded a 389 amino acid protein, which was localized to the nucleus, with obvious coiled coil areas. EsABI5 had high homology, with ABI5 proteins from *Hordeum vulgare*, *Triticum monococcum*, *Triticum aestivum*, and *Aegilops tauschii*. The conserved domains of EsABI5 belonged to the basic leucine zipper domain superfamily. EsABI5 had 10 functional interaction proteins with credibility greater than 0.7. *EsABI5* expression was upregulated in roots and leaves under NaCl stress and was upregulated in leaves and downregulated in roots under ABA treatment. Notably, tobacco plants overexpressing the *EsABI5* were more sensitive to salt stress, as confirmed by the determining of related physiological indicators. *EsABI5* expression affected the ABA and mitogen-activated protein kinase pathways. Therefore, *EsABI5* is involved in antisalt responses in these pathways and plays a negative regulatory role during salt stress.

## 1. Introduction

Abscisic acid (ABA) is a vital plant hormone that orchestrates plants in their adaptive response to abiotic stresses, such as salt, drought, and cold stresses, and regulates complicated metabolic and physiological mechanisms essential for survival in adverse environments [1,2,3,4]. ABA will quickly accumulate and cause stomatal closure to limit water loss through transpiration under abiotic stresses. In addition, ABA will mobilize a series of genes that can protect cells from ensuing oxidative damage due to prolonged stress, and the signaling network mediating these various responses against abiotic stresses is highly complex [5,6]. ABA is involved in complex signaling networks, including the PYR/PYL/RCAR ABA receptor, PP2C protein phosphatase, SnRK2 protein kinase, and ABI5/AREB/ABF transcription factor networks [7,8]. ABA-insensitive 5 (ABI5) is a member of the bZIP A subfamily and has been shown to regulate ABA signaling and stress-induced gene expression [9,10]. In the regulatory network of plants, ABI5 is activated by SNF1-related protein kinase 2 self-phosphorylation, binds to ABA response elements (ABREs) on the promoter of ABA-responsive genes, and regulates ABA-induced gene expression [11]. ABREs in the *ABI5* promoter region (between −1376 and −455 bp) are enriched by ABRE binding factor 3 (ABF3) in *Arabidopsis thaliana* under normal conditions and in the context of 150 mM NaCl salt stress [12]. The expression of ABI5 is regulated by ABF3 and can contribute to the salt tolerance of *Arabidopsis thaliana* [13]. The ubiquitin 26S proteasome (26SP) inhibits degradation of the ABI5 binding protein to stabilize the activity of ABI5 [14], thereby resulting in the regulation of ABI5 expression in the nucleus and cytoplasm and subsequently affecting the activity of the ABA signaling pathway [15,16]. The 26SP system can effectively degrade many key regulatory factors involved in plant development, and the β5 subunits of the 26SP system are essential for promoting plant growth under salt stress. PBE1 is a β5 subunit that regulates the ABI5 protein and modulates the expression of several downstream genes under stress conditions [17]. ABI5 mediates many important physiological processes of plants, including late embryonic development, adverse stress reactions, and plant growth/tolerance in response to abiotic stress [18,19,20,21,22]. *OsABI5*, which has been isolated from the panicle of *Oryza sativa* L., is a bZIP transcription factor involved in rice fertility and stress tolerance; the expression of *OsABI5* is induced by ABA and high salinity and suppressed by drought and cold (4 °C) in seedlings, while its overexpression in rice confers high sensitivity to salt stress [2]. Barley (*Hordeum vulgare*)-derived ABI5 is involved in the ABA-dependent drought response [22], and Chinese cabbage (*Brassica rape*)-derived ABI5s, i.e., BrABI5a and BrABI5b, are significantly induced by ABA, acting as positive modulators of ABA signaling [23]. The expression of *ZmABI5*, which has been isolated from maize (*Zea mays* L.), is upregulated in maize leaves and downregulated in maize roots under NaCl and ABA treatments; additionally, when transgenic tobacco plants overexpressing *ZmABI5* were treated with NaCl, mannitol, or high and low temperatures, they displayed obvious stress-sensitive phenotypes, and they play important negative regulatory roles in stress responses [24].

Siberian wildrye (*Elymus sibiricus* L.) is a typical species of the genus *Elymus*, which has resistance to cold, drought, and salt, and can grow well in barren, highly saline, or high humus soils, making it one of the main planting grasses in northern China. Siberian wildrye is also widely utilized in animal feed and is important for ecological environment protection. Moreover, this plant is an important source of stress resistance genes for wheat family-related crops and is commonly used for natural grassland grazing, artificial grass planting, and degraded grassland restoration, making it an excellent grass for ecological governance and animal husbandry [25,26,27].

Siberian wildrye is a salt-tolerant and high-quality forage; it not only has large potency in saline–alkali management but is also an important resistant gene source for Triticeae crops. The ABA signaling pathway plays a very vital role for plants in response to salt stress, and *ABI5* is an important transcription factor in the ABA signal transduction pathway. To date, few *ABI5* have been characterized in Siberian wildrye. Herein, we focused on elucidating the role of *ABI5* from Siberian wildrye (*EsABI5*) in ABA signaling during salt stress responses. Our findings clearly indicate that *EsABI5* is involved in ABA-dependent salt responses, which can provide a theoretical basis for the study of salt-tolerant regulation, molecular mechanism, and breeding of Siberian wildrye.

## 2. Results

### 2.1. Isolation and Basic Analysis of EsABI5

In this study, *EsABI5* was isolated from Siberian wildrye and had a total length of 1563 bp, including a 1170 bp open reading frame, and it encoded a protein of 389 amino acids. The molecular formula of EsABI5 is C_1747_H_2812_N_528_O_561_S_34_, including 5682 atoms, and its molecular mass is 41,278.79 Da. The theoretical isoelectric point was 5.74, and the EsABI5 protein showed nuclear localization. The protein residues in the 300–382 amino acid region (windows 14, 21, and 28) showed different coiled coil structures (Figure 1).

Three related domains with low E values were obtained at positions 304–364 in the coiled coil; these domains included bZIP_plant_bZIP46, basic region leucine zipper (BRLZ), and BZIP_1, indicating that the protein belongs to the bZIP transcription factor family (Figure 2, Table 1).

We screened out nine sequences that had homology with the EsABI5 protein by BLASTp and performed phylogenetic tree analysis (Figure 3). EsABI5 was most homologous with HvABI5 (*Hordeum vulgare*), TmABI5 (*Triticum monococcum*), TaABI5 (*Triticum aestivum*), TuABI5 (*Triticum uratu*), and AtaABI5 (*Aegilops tauschii*), showing high homology and close relationships. Lower homology and distant relationships were observed with OsABI5 (*Oryza sativa*), ZmABI5 (*Zea mays*), AtABI5 (*Arabidopsis thaliana*), and SvABI5 (*Setariaviridis)*.

We found 10 proteins that interacted with EsABI5 (credibility score > 0.7; Figure 4). The B3 domain containing the protein VP1 (VP1B3) had a credibility score of 0.916, which is the best functional partner of the EsABI5 protein. Two of these proteins were uncharacterized (UP); the others were ABA-inducible protein kinase (ABAIPK), CBL-interacting protein kinase 22 (CIPK22), WD repeat-containing protein DWA1 (DWA1WD), CBL-interacting protein kinase 29 (CIPK29), E3 ubiquitin-protein ligase KEG (KEGE3), serine/threonine-protein kinase SAPK8 (SAPK8), and serine/threonine-protein kinase SAPK3 (SAPK3). All of these functional interacting proteins play regulatory roles in plant stress.

### 2.2. Subcellular Localization of EsABI5

We examined the subcellular location of the EsABI5 protein. GFP-fused EsABI5 was mainly localized to the nucleus of the cells (Figure 5). These data indicate that EsABI5 is a nuclear protein, and imply that EsABI5 is a transcription factor.

### 2.3. Expression of EsABI5 under Salt and ABA Treatment

Under salt stress, the expression of *EsABI*5 increased in both leaves and roots, but the degree of upregulation in leaves (approximately three times that of the control) was greater than that in roots (approximately 1.5 times that of the control). Under ABA treatment, the expression of *EsABI5* in leaves first increased and then decreased. The expression of *EsABI5* was significantly different under different ABA concentrations (*p* < 0.05). When the ABA concentration was 50 µM, the expression of *EsABI5* was significantly higher than that of other treatments (*p* < 0.05) and approximately seven times higher than that of the control (Figure 6). The expression of *EsABI5* was significantly downregulated in roots (*p* < 0.05), but there were no significant differences under different ABA concentrations (*p* > 0.05). When the ABA concentration was 50 µM, the expression of *EsABI5* was 41 times lower than that of the control (Figure 7). Thus, *EsABI5* is involved in salt stress and ABA hormone regulation, and the roots may be the key antisalt organ of Siberian wildrye.

### 2.4. Decreased Salt Tolerance of EsABI5 Transgenic Tobacco

OT became more yellow and wilted compared with WT plants under salt stress (Figure 8), demonstrating that transgenic tobacco is more sensitive to salt stress. This result was further confirmed by evaluating the related physiological indicators (Figure 9).

Under salt stress, the malondialdehyde (MDA) content of transgenic tobacco (OT) was significantly higher than that of WT tobacco (*p* < 0.05). Superoxide dismutase (SOD) activity in OT was significantly lower than that in WT tobacco (*p* < 0.05). The soluble sugar content of OT was significantly higher than that of WT tobacco (*p* < 0.05). Additionally, under salt stress, the soluble sugar content of OT decreased, whereas that of WT tobacco increased. The proline content of OT was significantly lower than that of WT tobacco (*p* < 0.05). Taken together, the results of the phenotypic and physiological indicators show that *EsABI5* plays a negative regulatory role during salt stress.

### 2.5. Involvement of EsABI5 in ABA and MAPK Signaling Pathways in Response to Salt Stress

Under salt stress, the ABA content of cells increases rapidly, after which ABA binds to PYR/PYL/RCAR receptors and triggers a second messenger Ca^2+^ signaling system. These bind to PP2C to form an ABA-PYR/PYL-PP2C complex that releases the SnRK2 protein kinase, which is inhibited by PP2C. In addition, SnRK2 can activate the downstream transcription factor ABI5 and MAPKKK-MAPKK-MAPK cascade reactions. ABI5 can lead to stomatal closure and, thus, affect the salt tolerance of plants. The MAPKKK-MAPKK-MAPK cascade can directly or indirectly participate in the salt tolerance of plants (Figure 10a).

To elucidate the mechanisms through which *EsABI5* overexpression decreases salt tolerance in OT, we analyzed the expression of 15 genes related to the ABA and MAPK pathways (Figure 10b). The expression levels of eight of these genes, including *PYL4*, *PYL9*, *CPK17*, *ABI1*, *MEK1*, *MKK3*, and *MPK7*, were significantly lower in WT tobacco than in OT (*p* < 0.05). Additionally, the expression levels of four genes, namely *ABI2*, *SAPK3*, *MPK4*, and *MPK17*, were lower in WT tobacco than in OT, although the differences were not significant (*p* > 0.05). Furthermore, the expression levels of *SAPK2* and *MAPKKK17* were significantly higher in WT tobacco than in OT (*p* < 0.05), and the expression levels of *SAPK9* and *SAPK10* were higher in WT tobacco than in OT, although the differences were not significant (*p* > 0.05). Expression of all genes except *MPK17* were downregulated in WT tobacco, and expression of all genes except *SAPK3*, *MEK1*, *MKK3*, *MPK7*, and *CPK17* were downregulated in OT tobacco under salt stress. There were complex regulatory relationships between *EsABI5* and these genes, and overexpression of *EsABI5* affected the ABA and MAPK pathways. Therefore, *EsABI5* is involved in antisalt responses in the ABA signal transduction pathway and MAPK cascade pathway and plays a negative regulatory role during salt stress.

## 3. Discussion

In this study, we isolated and evaluated *EsABI5* from Siberian wildrye for the first time, making it possible to initiate studies into the role of *EsABI5* in the regulation of ABA and salt responses in Siberian wildrye. EsABI5 showed high similarity to HvABI5, TmABI5 (*Triticum monococcum*), TaABI5 (*Triticum aestivum*), and AtABI5 (*Aegilops tauschii*). Subcellular localization showed that the EsABI5 protein was mainly localized in the nucleus. Moreover, EsABI5 contained the structural proteins bZIP46, CBLZ, and bZIP1, which belong to the bZIP superfamily. Therefore, we conclude that EsABI5 belongs to the bZIP transcription factor family, consistent with the results for OsABI5 [2], HvABI5 [22], and ZmABI5 [24]. Members of the bZIP family are involved in the ABA signaling pathway, which can be fully activated by ABA. As a regulator of ABA signaling and stress tolerance, bZIP plays an important role in ABA responses [3,28,29,30,31,32].

EsABI5 had 10 functional interaction proteins. We found a high correlation between VP1B3 and EsABI5. The B3 domain is a highly conserved domain that is present in many higher plant genomes and can bind to DNA. The B3 domain of the VP1 protein has sequence-specific DNA binding activity that can play a regulatory role in plant growth, development, and stress by binding to specific DNA [33]. Calcium (Ca^2+^) is an essential nutrient in eukaryotes and signaling mediums. In plants, calcineurin B-like proteins (CBLs) are a unique set of Ca^2+^ sensors that unlock Ca^2+^ signals by activating a series of plant-specific protein kinases (CIPKs) [34]. CBLs are involved in decoding calcium signals, have no kinase activity on their own, and can only transmit signals downstream through CIPK interactions [35,36]. CBL/CIPK interactions are also involved in ABA-dependent signaling processes [37]. CIPK is a Ca-dependent, CBL-specific serine/threonine (Ser/Thr) targeting protein kinase belonging to the sucrose nonfermenting-related protein kinase (SnRK3) family, which interacts with the CBL protein and plays a key role in sensing calcium signals and transmitting stress response signals [38]_._ SAPK3 and SAPK8 are Ser/Thr SnRK2 protein kinases and members of the SnRK2 family closely related to plant resistance; overexpression of *SnRK2* activates the expression of a downstream series of resistor-related genes, thus improving plant resistance [39,40]. KEG is an important RING-type E3 that negatively regulates ABA signal transduction by targeting ABI5 for ubiquitination and degradation. ABI5 is degraded following ubiquitination by KEG [41,42]. When plants detect ABA, they promote the self-ubiquitination and degradation of KEG, thereby reducing KEG-mediated ABI5 degradation and promoting the ABA signal response [43]. The MAPK cascade is an important signal transduction pathway in plants that respond to salt stress. Notably, KEG degrades and regulates key genes in the MAPK cascade by ubiquitinating MKK4 and MKK5 proteins [44]. ABA exposure and adverse stress can activate CIPK26, which can phosphorylate KEG and ABI5, activate the transcriptional activity of ABI5, and promote KEG self-ubiquitination and degradation, further leading to a reduction in KEG-mediated CIPK26 and ABI5 degradation and, thus, enhancing plant responses to ABA [45]. The WD protein is involved in the biological processes of various cells and organisms and plays an important role in the regulation and development of plant growth under adverse conditions, such as high salt, drought, and low temperature [46,47,48]. DWA1 is involved in regulating the level of jasmonic acid, which plays an important regulatory role in the growth and development of plants and is necessary for plants to resist stress and complete their growth/development and life cycle [49,50]. The WD40 replication protein (WD40 protein), a universal scaffold for protein interactions, is involved in various biological processes such as plant stress and hormone responses. The WD40 protein 1 (XIW1) interacts with Xpo1 following exposure to ABA in the nucleus. Moreover, XIW1 interacts with ABI5 to maintain its stability. Under salt stress, mutations in XIW1 reduce the induction of ABI5 and ABA response genes [51]. Overall, these results support the involvement of EsABI5 in the ABA signaling pathway and in plant adaptation to stress.

Salt stress can regulate *ABI5* expression [52]. Under salt stress, the expression of *AtABI5* in *Arabidopsis* leaves is upregulated, but not significantly; however, significant upregulation has been observed in roots [53,54]. Salt resistance and sensitivity in sorghum (*Sorghum bicolor*) are not obviously altered in the leaves following changes in *SbABI5* expression; however, increases have been observed in the roots of *SbABI5* salt-tolerant germplasm, consistent with observations in *Arabidopsis thaliana*. Thus, *SbABI5* may be the key salt resistance-related gene in the *SbABI**s* family, and root *SbABI5* expression may be a critical parameter in the ABA signal transduction pathway modulating sorghum salt resistance. After 20 µM ABA treatment, *SbABI5* expression was significantly upregulated in the roots of the two germplasms, whereas after 100 µM ABA treatment, *SbABI5* expression was not significantly altered, indicating that a high concentration of ABA could inhibit the expression of *SbABI5* [54]. In this study, the expression of *EsABI5* was upregulated in both roots and leaves under NaCl stress; however, the extent of upregulation in the leaves (3-fold) was higher than that in the roots (1.5-fold). Under ABA treatment, *EsABI5* was upregulated in the leaves but was downregulated in the roots. *EsABI5* expression in the roots was not as sensitive to ABA concentration changes as that in the leaves, but the extent of downregulation in the roots (41-fold) was higher than the extent of upregulation in the leaves (7-fold). Therefore, these findings support that *EsABI5* is related to salt stress and the ABA regulation mechanism. The results of this study also show that with an increase in ABA concentration, the expression of leaves *EsABI5* increases and then decreases; this is opposite for roots *EsABI5*, indicating that ABA plays a positive regulatory role within a certain concentration range and may play a negative regulatory role beyond this concentration. This result is similar to Gietler’s [55].

The physiological and molecular mechanisms through which plants respond to salt stress are complex. Growth and development, hormones, metabolism, osmotic adjustment, membrane protective material and active oxygen balance, salt stress proteins, and plant salt stress information transmission mechanisms in the body can change via modulation of related gene expression; however, the specific functions of most of the genes related to these factors are unknown [56,57]. Due to tobacco’s short growth cycle, mature transgenic technology, and simple and easy operation of the transgenic process, tobacco is often used as experimental material for the identification of exogenous functional genes, which speeds up the research progress of functional genes. In this study, we evaluated the functions of *EsABI5*, a core transcription factor in the ABA signaling system in Siberian wildrye under salt stress. The amplified CDS fragment of *EsABI5* was successfully introduced into the vector pART-CAM, and the tobacco leaf disk was infected by the *Agrobacterium*-mediated method. The buds were then induced to differentiate, and after about 6 months of tissue culture, positive lines were obtained. After polymerase chain reaction (PCR) and RT-qPCR detection of the tobacco plants overexpressing, the *EsABI5* (OT) showed that the morphology of OT was significantly altered. For example, the plant height, roots, and leaves of transgenic plants were larger than those of WT plants, consistent with the involvement of *ABI5* in plant seedling growth [29]. OT and WT tobacco were then exposed to salt stress, and the results showed that transgenic plants were more sensitive to salt stress. This result was further confirmed by evaluation of related physiological indicators.

The ABA signal transduction pathway and MAPK cascade pathway are important pathways regulating plant responses to salt stress. The MAPK cascade pathway can directly or indirectly participate in the ABA signal transduction pathway, and the ABA signal pathway can also regulate the expression of related genes in the MAPK cascade pathway [58,59]. The MEKK1/MKK1/MKK2/MPK4 axis of the MAPK signaling pathway exists in *Arabidopsis thaliana*, and *mekk1* mutants show tolerance to salt [60]. *MAPKK1* [61], *MPK4*, *MPK6* [62], *MKK9*, *MPK3* [63,64], and *OsMAPK44* [65] in rice can all be activated by salt. Additionally, *GhMPK7* in cotton [66] and *ZmMPK3* in maize [67] are involved in the response to salt stress. In this study, among 15 genes related to ABA and MAPK pathways, the expression of 11 genes was lower in WT tobacco than in OT, whereas the expression of four genes was higher in WT tobacco than in OT. Under salt stress, all genes except *MPK17* were downregulated in WT tobacco, whereas all genes except *SAPK3*, *MKK3*, *MAPKKK17*, *MEK1*, and *CPK17* were downregulated in OT. There are complex regulatory relationships between *EsABI5* and these genes, and overexpression of *EsABI5* affected the expression of genes related to the ABA and MAPK pathways. Yan [24] speculated that *ZmABI*5 might affect the MAPK cascade, consistent with the results of this study.

Overall, these results show that *EsABI5* is involved in antisalt responses in the ABA signal transduction pathway and MAPK cascade pathway and acts as a negative regulator. These findings establish a basis for further studies into the mechanism of ABA signaling in Siberian wildrye and salt-tolerant regulation in Siberian wildrye and related plants. In our subsequent studies, we will evaluate the mechanisms of *EsABI5* under other abiotic stresses and hormonal treatments.

## 4. Materials and Methods

### 4.1. Materials

High-salinity Siberian wildrye (Tuzuo Banner of Hohhot City, Inner Mongolia, China, E 111°27′, N 40°47′, H 1157 m) was cultivated to obtain seedlings for antisalt experiments. Uniform, plump seeds were selected, and the lemma were removed using an ultra-clean workbench. Samples were washed with distilled water once, sterilized with NaClO for 15 min, rinsed with distilled water four times, and placed in Petri dishes (50 seeds/dish). After seeds grew to 2–3 cm, the buds were planted in double disks. When the seedlings cultured with distilled water had grown to approximately 5–6 cm, Hoagland nutrient solution was added and replaced with distilled water every 3 days. Plants were grown in a 25 °C light incubator with 37% humidity, 14-h light/10-h dark cycle, and 164 μM·m^−2^·s^−1^ light intensity. The experiments were repeated three times. When the second true leaf emerged, some of the samples were used as controls, some were placed in 250 mM NaCl solution, and some were placed in 20, 50, or 100 µM exogenous ABA (cat. no. CA1011, Coolaber, Beijing, China) solution. Assessments of biological processes were repeated three times. When the phenotype changed, the leaves and roots of 5 different treatments were single collected, quickly frozen in liquid nitrogen, and stored at −80 °C.

### 4.2. Primer Design

Premier 5.0 software was used to identify the homologous region of the *ABI5* gene of related species, including *Triticum aestivum* (KX002276.1, AF519804), *T. monococcum* (AB286054.1), and *Aegilops tauschii* subsp. *Tauschii* (XM_020327051), and to design the primers. All primer sequences are shown in Table 2 and Table 3, and all primers were synthesized by Invitrogen (Carlsbad, CA, USA).

### 4.3. Reverse Transcription Quantitative PCR, PCR Product Purification, and Clone Sequencing

TRIzol Universal Total RNA Extraction Reagent (cat. no. DP405; Tiangen, Beijing, China) was used to extract RNA from samples. A PrimeScript 1st Strand cDNA Synthesis Kit, 3′-Full RACE Core Set with PrimeScriptRTase, and Smarter RACE 5′/3′ Kit were used to synthesize cDNA (TaKaRa, Dalian, China). TaKaRaTksGflex DNA Polymerase was used for PCR. A TakaRaMiniBEST Agarose Gel DNA Extraction Kit ver. 4.0 was used to recover and purify the PCR product bands. The purified products were treated with a DNA A-Tailing Kit. Ligase from TaKaRa DNA Ligation Kit ver. 2.1 was used for ligation with the T-Vector pMDTM20. The plasmid was heat transferred into *Escherichia coli* competent cells (JM109), and cells were cultured overnight at 37 °C. Positive single clones were selected and designated CTG0957-PCR-1 and CTG0957-PCR-2. The primers M13-47 and RV-M were used for sequencing.

### 4.4. Bioinformatics Analysis of EsABI5

The obtained open reading frame base sequence of *EsABI5* was translated into an amino acid sequence using Bioedit V7.0. The amino acid sequence was imported into the NCBI BLAST database for BLASTp analysis against nine amino acid sequences showing homology with EsABI5 protein. Phylogenetic trees were generated using Mega 7.0 with the neighbor-joining method, and the phylogenetic relationships were analyzed. NCBI was used to predict EsABI5 domains, and PROSITE (https://prosite.expasy.org/, accessed on 10 March 2020) was used to generate images. String (https://string-db.org/, accessed on 10 March 2020) was used to analyze the protein interaction network, and Cytoscape 3.7.1 was used to plot the protein interaction network. All default settings were used.

### 4.5. Subcellular Localization

*Trelief*^TM^SoSoo Cloning Kit Ver. 2 (cat. no. TSV-S2, Tsingke, Beijing, China) was used to clone the CDS of EsABI5, and pCAMBIA1302 was the vector. The pCAMBIA1302-EsABI5 plasmid was introduced into the *Agrobacterium tumefaciens* strain GV3101; other parts of this method were similar to Bai’s [23].

### 4.6. Real-Time Fluorescence Quantitative PCR

The reaction mixture (total volume 20 μL) included 10 μL SYBR Premix Ex Taq (2×), 1 μL forward primer (10 μM), 1 μL reverse primer (10 μM), 2 µg cDNA, and ddH_2_O. The two-step fluorescence quantitative PCR amplification conditions were as follows: 95 °C for 5 min; 40 cycles of 95 °C for 10 s and 60 °C for 30 s; detection at 72 °C; and a dissolution curve of 95 °C for 15 s, 60 °C for 60 s, and 95 °C for 15 s, with continuous signal detection. *Gapdh* and *NbUbe3* served as the internal reference genes for normalizing the expression of *EsABI5* plants and WT, respectively [68,69]. The 2^−△△Ct^ method was used to calculate relative gene expression [70].

### 4.7. Overexpression Vector Construction, Tobacco Transformation, and Line Propagation

The amplified CDS of *EsABI5* was successfully constructed into the vector pART-CAM, which used the method of restriction enzyme double digestion. Tobacco leaf disks were infected using an *Agrobacterium*-mediated method, and the buds were induced to differentiate, yielding positive buds. Twelve transgenic lines were obtained by screening resistant culture mediums and by PCR detection using specific primers. Transgenic and WT tobacco seedlings with consistent growth were selected and transplanted into flowerpots (sandy soil:nutrientsoil:vermiculite = 1:1:0.5) at 25 °C with 60% humidity, 14-h/10-h light/dark cycle, and 61.64 μM·m^−2^·s^−1^ light intensity. After normal watering and culture for 2 weeks, 4 transgenic lines were detected by PT-qPCR (primers were EsABI5-F1 and EsABI5-R1). Then, transgenic and WT tobacco were irrigated with 250 mM NaCl solution, weighing water daily. Distilled water was used as the control group.

### 4.8. Physiological Indexes and Data Analyses

MDA content and SOD activity were determined by malondialdehyde (MDA) (cat. no. BC0025; Solarbio, Beijing, China) assay kits and superoxide dismutase (SOD) (cat. No. WFY1; Cominbio, Suzhou, China) assay kits, respectively. Soluble sugar content was determined by anthrone colorimetry. Proline content was determined by ninhydrin colorimetry.

Data were collated using Microsoft Excel 2016 software and plotted using SigmaPlot 12.5 software and Adobe Illustrator CC 2018. SPSS 20.0 software was used for variance analysis (Duncan method, *p* < 0.05).

## Figures and Tables

**Figure 1 plants-10-01351-f001:**
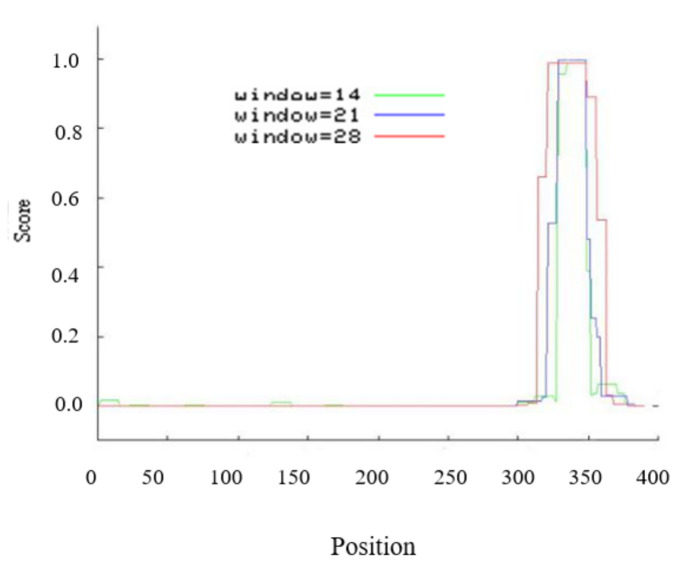
Coil analysis of EsABI5 protein.

**Figure 2 plants-10-01351-f002:**
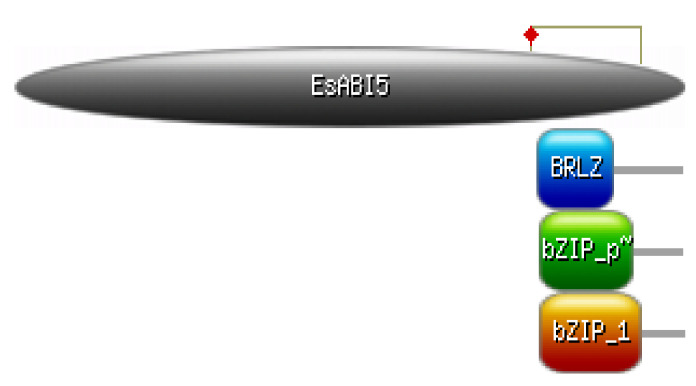
Domain structure of EsABI5 protein.

**Figure 3 plants-10-01351-f003:**
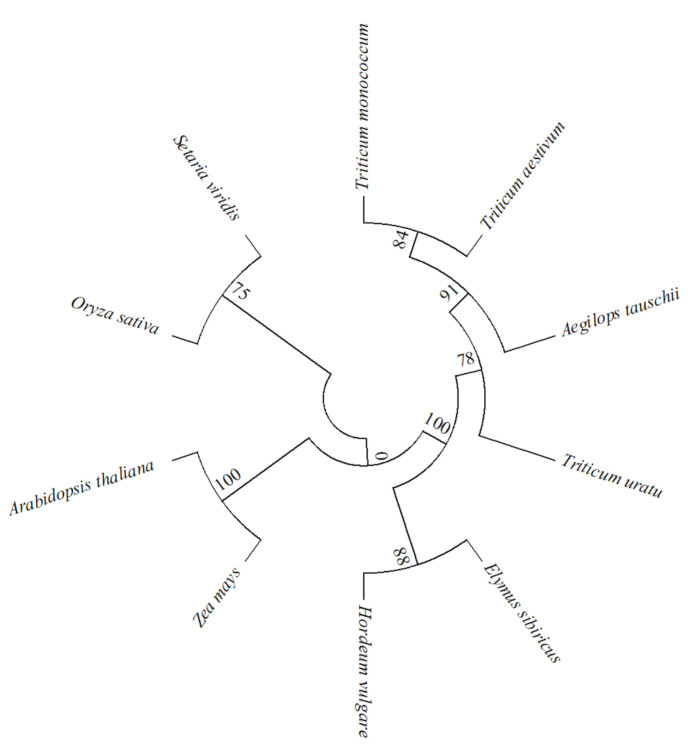
Genetic relationship analysis based on EsABI5 protein.

**Figure 4 plants-10-01351-f004:**
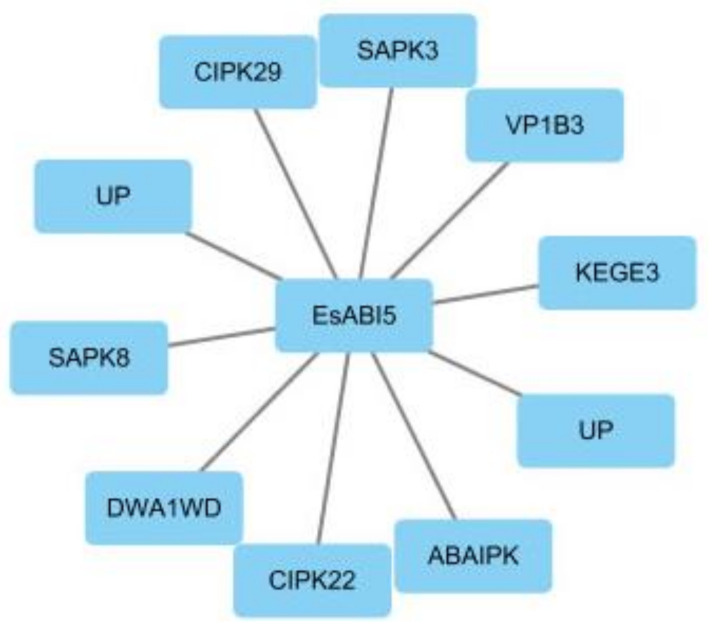
Analysis of proteins interacting with EsABI5 protein.

**Figure 5 plants-10-01351-f005:**
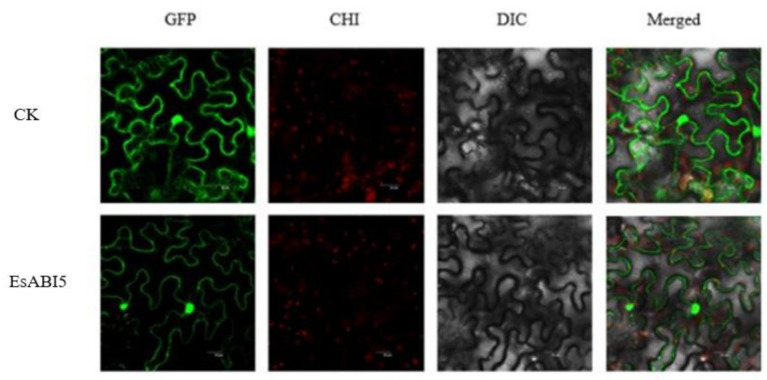
Subcellular localization of EsABI5 protein in tobacco epidermal cells: GFP, green fluorescence protein under dark field; CHI, chloroplast autofluorescence; DIC, cell morphology of the lower epidermis of a tobacco leaf under bright field. Merged: overlay of GFP, CHI, and DIC.

**Figure 6 plants-10-01351-f006:**
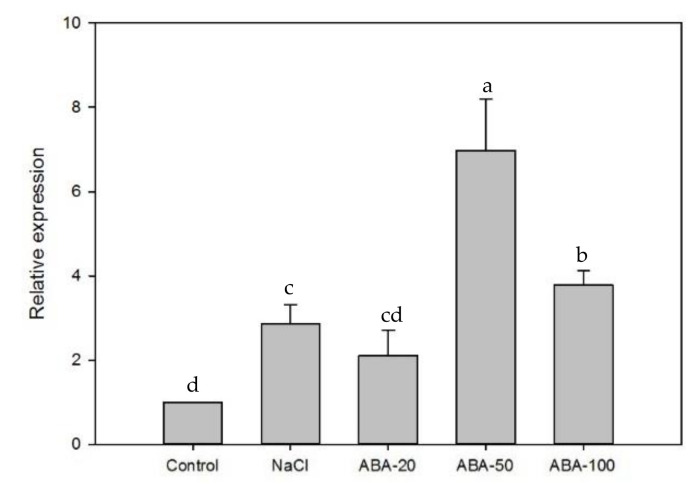
*EsABI5* expression under different treatments in leaves. Different letters indicate significant differences compared to the control group (*p* < 0.05). ABA-20 indicates that the concentration of exogenous ABA is 20 µmol/L. ABA-50 indicates that the concentration of exogenous ABA is 50 µmol/L. ABA-100 indicates that the concentration of exogenous ABA is 100 µmol/L.

**Figure 7 plants-10-01351-f007:**
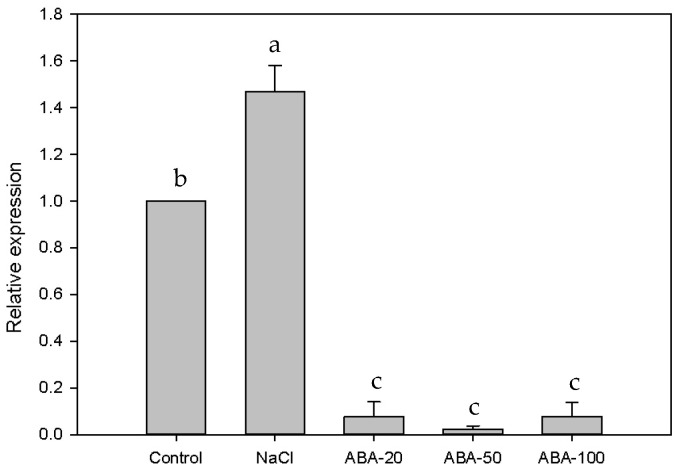
*EsABI5* expression under different treatments in roots. Different letters indicate significant differences compared to the control group (*p* < 0.05). ABA-20 indicates that the concentration of exogenous ABA is 20 µmol/L. ABA-50 indicates that the concentration of exogenous ABA is 50 µmol/L. ABA-100 indicates that the concentration of exogenous ABA is 100 µmol/L.

**Figure 8 plants-10-01351-f008:**
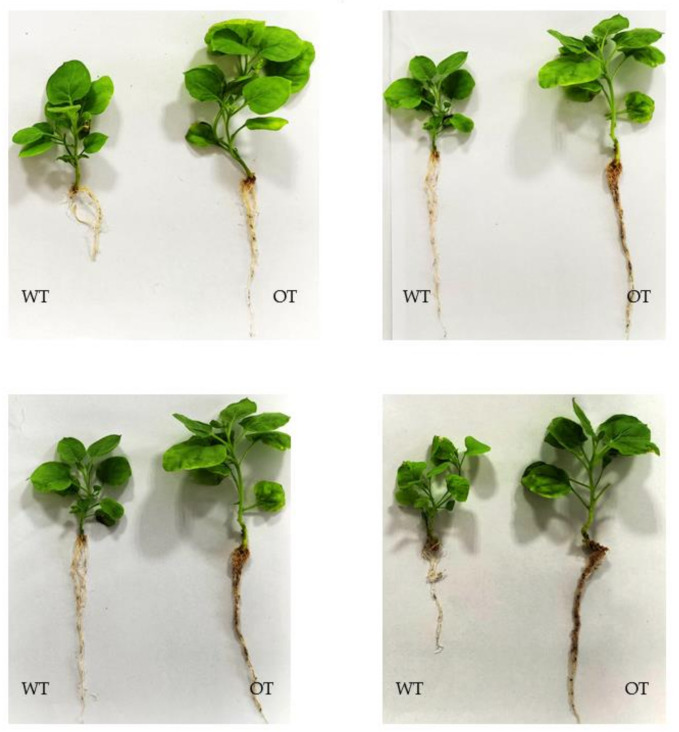
Effects of salt stress on the phenotypes of wild-type and transgenic tobacco.

**Figure 9 plants-10-01351-f009:**
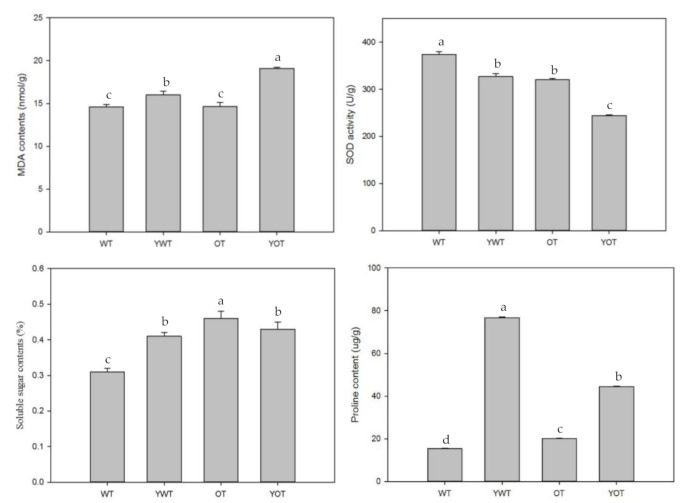
Effect of different treatments on physiological indexes of tobacco: WT, wild-type tobacco; YWT, salt-stressed wild-type tobacco; OT, transgenic tobacco; YOT, salt-stressed transgenic tobacco. Different letters indicate significant differences compared to the control group (*p* < 0.05).

**Figure 10 plants-10-01351-f010:**
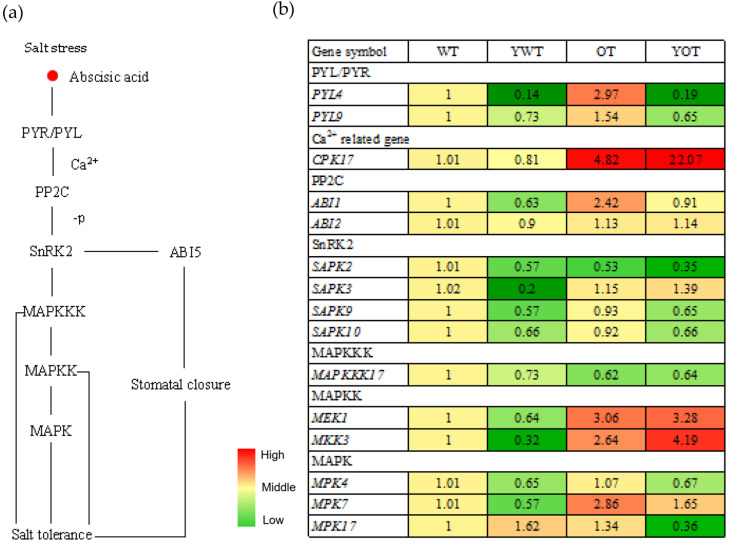
Expression analysis of genes involved in ABA and MAPK pathways: (**a**) the roadmap of ABA and MAPK pathways; (**b**) heat map of ABA and MAPK pathways-related gene expression; WT, wild-type tobacco; YWT, salt-stressed wild-type tobacco; OT, transgenic tobacco; YOT, salt-stressed transgenic tobacco.

**Table 1 plants-10-01351-t001:** Domain prediction of *EsABI5*-encoded proteins.

Name	Position Interval	E Value
bZIP_plant_BZIP46	305–359	3.67 × 10^−23^
BRLZ	304–348	3.09 × 10^−14^
bZIP_1	305–364	1.44 × 10^−10^

**Table 2 plants-10-01351-t002:** Primer sequences used for cloning.

Name	Sequence 5′→3′	Application
CTG0957 F1	GCGGCAGTCTTCCATCTTCG	Homologous gene acquisition
CTG0957 R1	TCCTCACCTTCGGCAACG	Homologous gene acquisition
CTG0957-1 F3	ACAGATGAACCCCGCGCAGCAGG	3′ RACE
CTG0957-1 F5	TGACGCAGGCTGACATGATGAACT	3′ RACE
CTG0957-1 F9	GATGATGGAACAGTCCAAGG	3′ RACE
CTG0957-2 R4	GAACATGCCGTTGGCCGGTGCCATC	5′ RACE
CTG0957-2 R6	CATGGACCATGCCGACCGGCACAG	5′ RACE
CTG0957-2 YZF1	CACAAGGCAAGCATATCGAG	RACE verification
CTG0957-1 YZR1	CCGCTCCGAAATGATAAGGT	RACE verification
CTG0957-1 YZR2	GTTATGATAGCTGAATGGCA	RACE verification
XhoI-EsABI5-F	CCGCTCGAGATGGCGTCGGCGATGAGCAA	EsABI5 CDS cloning
XbaI-EsABI5-R	GCTCTAGATCACACGTGGTGGTGGTGGT	EsABI5 CDS cloning

**Table 3 plants-10-01351-t003:** Primer sequences used for RT-qPCR.

Gene Symbol	GenBank Accession Number	Primer Name	Sequence (5′→ 3′)
*NbUbe35*	SRP118889	NbUbe35-F	CTTCAGATTCGCACCGTTCT
		NbUbe35-R	CCAATGCTTCGCAATGTTCTC
*Gapdh*	AF251217.1	Gapdh-F	ATGAGGACCTTGTTTCCACTGACTT
		Gapdh-R	GTGCTGTATCCCCACTCGTTGT
*EsABI5*	MN607227.1	EsABI5-F	ACGCAGGCTGACATGATGA
		EsABI5-R	CGGCTGACTCACGGTTCTT
*EsABI5*	MN607227.1	EsABI5-F1	CGGCAGTCTTCCATCTTCG
		EsABI5-R1	GGAACTCCTCGGCATTCCAG
*PYL4*	XM_019392528.1	PYL4-F	TCCGCGTTGTTTCTGGC
		PYL4-R	GTTTCTTCCTTAGTATTCCCTTGTG
*PYL9*	XM_019409396.1	PYL9-F	GTTTGGTCATTAGTGAGGAGGTTT
		PYL9-R	TTGCTTGTGGTAGCTGGGAG
*ABI1*	NM_001336190.1	ABI1-F	CGCCTCTTGTGACCTTGCT
		ABI1-R	CCGCTTTCGGGTTCTGTTA
*ABI2*	XM_019393623.1	ABI2-F	GGTAGGAGGGCTTGGTAGTGA
		ABI2-R	TGGACAACGGCATGGGTA
*SAPK2*	XM_033652288.1	SNF1-F	AGTGGCAAGGCTTATGAGGG
		SNF1-R	CTCTTTGAATCTGACTATGTTAGGGTG
*SAPK3*	XM_019398597.1	SAPK3-F	CAAAGGAGCTTGTTGCTGTCA
		SAPK3-R	GAGCCTCATCTTCACTAAATCTACC
*SAPK9*	EH367211.1	SAPK9-F	ATTCTACTCGACGGAAGTGCTG
		SAPK9-R	GATCGCTTGAATCTTGAAATGG
*SAPK10*	XM_019398090.1	SRK2E-F	AGTGGCAAGGCTTATGAGGG
		SRK2E-R	CTCTTTGAATCTGACTATGTTAGGGTG
*MAPKKK17*	NM_001325618.1	NPK1-F	TCCTGGTGGCTCAATCTCG
		NPK1-R	GTCAACAAGTATGTTTGCTCCCT
*MPK4*	XM_019391189.1	MPK4-F	GTCGTAACACGGTGGTATCGG
		MPK4-R	CTGGCATCATCAGGTGATCCTAT
*MPK7*	XM_019380635.1	MPK7-F	CGAATAGAATTGATGCGCTGAG
		MPK7-R	GGATAAAGGCTGCGACGACT
*MPK17*	NM_001325663.1	MPK17-F	TCCCATCTGCCAGTGAGGTT
		MPK17-R	TTGCTGCGAGGCTTTGAGT
*MKK3*	XM_019380228.1	MKK3-F	TTTCTCACCTGCCTCTACATCG
		MKK3-R	ACACTACTTGCACCGCTACCTAT
*MEK1*	NM_001326016.1	MEK1-F	CTTTCACGACGGCGATTTAC
		MEK1-R	GAACAACACCACCACTTCCCT
*CPK17*	XM_019401998.1	CPK17-F	AGGATGGAGAAGCACCAGATACAC
		CPK17-R	CACCCTGCAATTACCCGAAG

## Data Availability

The sequence for *EsABI5* was deposited in GenBank (accession number: MN607227.1).

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
