# Peer review of "Siberian Wildrye (Elymus sibiricus L.) Abscisic Acid-Insensitive 5 Gene Is Involved in Abscisic Acid-Dependent Salt Response"

_plants, 2021, doi:10.3390/plants10071351_

Round 1
Reviewer 1 Report
The manuscript deals with a relevant subject (Abscisic Acid-Insensitive 5 Gene is Involved in Abscisic Acid-Dependent Salt Response) to PLANTS. The ms is very interesting, well written, with an interesting set of well-presented results and adequate discussion, and the appropriate topics are supported by the literature. I recommend that the manuscript should be accepted after minor revision.
Minor points:
- Abstract: Authors should put EsABI5 in full in the first time.
- Material and methods: Authors should use photosynthetic photon flux density units.
- Material and methods: authors should give information about how exogenous ABA solutions were applied and how they were prepared.
- Material and methods: authors should change the sentence: “MDA content and SOD activity were determined by micro malondialdehyde (MDA)”. Micro malondialdehyde or micro assay for malondialdehyde?
Author Response
- Abstract: Authors should put EsABI5 in full in the first time.
I have modified it, which is as follows:
Elymus sibiricus L. abscisic acid-insensitive 5
- Material and methods: Authors should use photosynthetic photon flux density units.
I have modified it, which is as follows:
Plants were grown in a 25°C light incubator with 37% humidity, 14-h light/10-h dark cycle, and 164 μmol·m-2·s-1 light intensity.
Transgenic and WT tobacco seedlings with consistent growth were selected and transplanted into flowerpots (sandy soil:nutrientsoil:vermiculite = 1:1:0.5) at 25°C with 60% humidity, 14-h/10-h light/dark cycle, and 61.64 μmol·m-2·s-1 light intensity.
- Material and methods: authors should give information about how exogenous ABA solutions were applied and how they were prepared.
I have added it, which is as follows:
When the second true leaf emerged, some of the samples were used as controls, some were placed in 250 mM NaCl solution, and some were placed in 20, 50, or 100 µM exogenous ABA (cat.no.CA1011, Coolaber, Beijing, China) solution. For 20, 50 and 100 µM ABA solution, put in 5.286 mg, 13.215 mg, 26.43 mg ABA power were dissolved in 1 L distilled water, respectively.( This part was not added in my manuscript because the preparation methods of other drugs were not added in the manuscript)
- Material and methods: authors should change the sentence: “MDA content and SOD activity were determined by micro malondialdehyde (MDA)”. Micro malondialdehyde or micro assay for malondialdehyde?
I have modified it, which is as follows:
MDA content and SOD activity were determined bymalondialdehyde (MDA) (cat. no. BC0025; Solarbio, Beijing, China) assay kits and superoxide dismutase (SOD) (cat. No. WFY1; Cominbio, Suzhou, China) assay kits, respectively.
Reviewer 2 Report
The manuscript cannot be considered for publication at the present stage due to the following reasons.
My main reservation concerns the lack of connections between particular parts of the manuscript. One part must be the base or starting point to the next part. Now it looks like each paragraph in the Results is independent. Especially the connections between experiments on Siberian wildrye and tobacco should be more clearly presented. Additionally, although I am not an expert in English, the language must be corrected. Also, the aim of the research must be written again. Now it is too laconic. It must be more detailed. Moreover, now it is written that ‘…only a few ABI5s have been characterized…’ but it is not clear.
Other problems
- Keywords should be changed because the keywords should not be the same as words already used in a title.
- The legends of all figures and table headings must be considerably improved because all tables and figures must be self-explanatory. So the legends and headings must be remarkably more detailed, and all abbreviations must be explained there.
- Figure 4 does not present an ‘ANALYSIS of proteins interacting’. It presents only a graphical list of proteins, but not ANALYSIS. Additionally, all abbreviations must be explained.
- Figure 5 has the same legend as Figure 4!
- Figures 6 and 7. The abbreviations must be explained.
- Figures 8, 9, 10, and the text. Why are so extraordinary abbreviations used? YWT, YOT? Such untypical abbreviations do not help in understanding the message.
- Figure 9. It must be SOD activity, not SOD content.
- The legend of Figure 10 needs correction. What does ‘Overview of’ mean?
- Materials and methods. Light intensity must be expressed in SI units. Section 4.1, instead of Lx it must be μM light quantum m−2 s−1. Section 4.7, what does mean ‘4500 light intensity’?
Author Response
The manuscript cannot be considered for publication at the present stage due to the following reasons.
My main reservation concerns the lack of connections between particular parts of the manuscript. One part must be the base or starting point to the next part. Now it looks like each paragraph in the Results is independent. Especially the connections between experiments on Siberian wildrye and tobacco should be more clearly presented.
The relationships between the parts: First, we isolated the EsABI5 gene from Siberian wildrye leaves, making it possible to initiate studies on the role of EsABI5 in the regulation of ABA and salt responses in Siberian wildrye. We analyzed the EsABI5 protein and subcellular localization of EsABI5 to understand its relationship with ABA and abiotic stress. Next, to determine whether EsABI5 regulates salt response in an ABA-dependent manner, we analyzed the expression of EsABI5 in roots and leaves under salt and ABA treatments to describe the response of EsABI5 to salt and ABA. Then, we subjected EsABI5 overexpression in tobacco to salt stress to determine how EsABI5 regulates salt response. Finally, we investigated the ABA and mitogen-activated protein kinase (MAPK) signaling pathways in WT and tobacco overexpressing EsABI5, revealing the general regulatory pathway of EsABI5 in tobacco.
Due to tobacco’s short growth cycle, mature transgenic technology, simple and easy operation of transgenic process, tobacco is often used as experimental materials for the identification of exogenous functional genes, which speeds up the research progress of functional genes. In this study, EsABI5 gene was transferred into tobacco to further analyze its function.
Additionally, although I am not an expert in English, the language must be corrected. Also, the aim of the research must be written again. Now it is too laconic. It must be more detailed. Moreover, now it is written that ‘…only a few ABI5s have been characterized…’ but it is not clear.
I rewrote the aim of the research, which is as follows:
Siberian wildrye is a salt tolerant and high quality forage, not only has large potency in saline-alkali management, but also is an important resistant gene source for Triticeae crop. ABA signaling pathway plays a very vital role for plants in response to salt stress, ABI5 is an important transcription factor in ABA signal transduction pathway. To date, few ABI5 has been characterized in Siberian wildrye. Herein, we focused on elucidating the role of ABI5 from Siberian wildrye (EsABI5) in ABA signaling during salt stress responses. Our findings clearly indicated that EsABI5 was involved in ABA-dependent salt responses. It can provide heoretical basis for the study on salt tolerant regulation, molecular mechanism and breeding of Siberian wildrye.
If it's my English writing problem, I'm willing to get a professional retouching company to retouch it.
Other problems
Keywords should be changed because the keywords should not be the same as words already used in a title.
I have modified it, which is as follows:
Abscisic acid; Siberian wildrye; salt stress; EsABI5 gene; expression mechanism; functional identification
The legends of all figures and table headings must be considerably improved because all tables and figures must be self-explanatory. So the legends and headings must be remarkably more detailed, and all abbreviations must be explained there.
- Figure 4 does not present an ‘ANALYSIS of proteins interacting’. It presents only a graphical list of proteins, but not ANALYSIS. Additionally, all abbreviations must be explained.
I have modified it, which is as follows:
We found 10 proteins that interacted with EsABI5 (credibility score> 0.7; Figure 4). B3 domain containing protein VP1 (VP1B3) with credibility score is 0.916, which is the best functional partner of EsABI5 protein. Two of these proteins were uncharacterized (UP); the others were ABA-inducible protein kinase (ABAIPK), CBL-interacting protein kinase 22 (CIPK22), WD repeat containing protein DWA1 (DWA1WD), CBL-interacting protein kinase 29 (CIPK29), E3 ubiquitin protein ligase KEG (KEGE3), , serine/threonine-protein kinase SAPK8 (SAPK8), and serine/threonine-protein kinase SAPK3 (SAPK3). All of these functional interacting proteins play regulatory roles in plant stress.
- Figure 5 has the same legend as Figure 4!
- I have modified it, which is as follows:
Figure 5. Subcellular localization of EsABI5 protein in tobacco epidermal cells.
- Figures 6 and 7. The abbreviations must be explained.
I have added it, which is as follows:
ABA-20 indicates the concentration of exogenous ABA is 20 µmol/L. ABA-50 indicates the concentration of exogenous ABA is 50 µmol/L. ABA-100 indicates the concentration of exogenous ABA is 100 µmol/L
- Figures 8, 9, 10, and the text. Why are so extraordinary abbreviations used? YWT, YOT? Such untypical abbreviations do not help in understanding the message.
WT usually refers to wild-type, so we used YWT to represent wild-type under salt stress. OT usually refers to overexpression-type, so we used YOT to represent overexpression-type under salt stress.
- Figure 9. It must be SOD activity, not SOD content.
- I have modified it, which is as follows:
- The legend of Figure 10 needs correction. What does ‘Overview of’ mean?
I have modified and added it, which is as follows:
Figure 10. Expression analysis of genes involved in ABA and MAPK pathways (a) The roadmap of ABA and MAPK pathways.
Under salt stress, ABA content of cells increases rapidly, then ABA binds to receptors and triggers a second messenger Ca2+ signaling system. These bind to PP2C to form ABA-PYR/PYL-PP2C complex, releases SnRK2 protein kinase, which is inhibited by PP2C. And SnRK2 can activate the downstream transcription factor ABI5 and MAPKKK-MAPKK-MAPKK cascade reactions. ABI5 can lead to stomatal closure and thus affect the salt tolerance of plants. MAPKKK-MAPKK-MAPK cascade can directly or indirectly participate in the salt tolerance of plants (Figure 10a).
- Materials and methods. Light intensity must be expressed in SI units. Section 4.1, instead of Lx it must be μM light quantum m−2 s−1. Section 4.7, what does mean ‘4500 light intensity’?
- Plants were grown in a 25°C light incubator with 37% humidity, 14-h light/10-h
dark cycle, and 164 μmol·m-2·s-1 light intensity.
- Transgenic and WT tobacco seedlings with consistent growth were selected and
transplanted into flowerpots (sandy soil:nutrientsoil:vermiculite = 1:1:0.5) at 25°C with 60% humidity, 14-h/10-h light/dark cycle, and 61.64 μmol·m-2·s-1 light intensity.
Reviewer 3 Report
Comments to Author
The manuscript entitled with “Siberian Wildrye abscisic acid-insensitive 5 gene is involved in abscisic acid-dependent salt response” by De et al. (Manuscript ID: plants-1233617) described on the expression of EsABI5 under salt stress to identify the role of EsABI5. In general, the experiments and manuscript were well conceived and conducted. The results compared with appropriate controls and important finding is presented in results and figures. Appropriate literature cited and discussed all relevant results. The technique is translatable to other systems and this increases the value of the work. However, I am still wondering if EsABI5 is involved in ABA-dependent salt response. I think there is not enough evidence in the manuscript. There are, however, several points of interest for the readers of this journal. The manuscript handled with several data and provide EsABI5 expression analysis under salt stress. Therefore, I recommend this manuscript for the publication in Plants after incorporating major revision.
[Major comments]
- Even authors provided 10 proteins that interacted with EsABI5, these are selected by bioinformatics analysis. I am not sure that all 10 proteins actually interact with EsABI5. Authors need to validate the protein interactions with transcriptional activation in yeast cell directly or expression profiles by RT-PCR indirectly.
- I think there is no evidence for ABA-dependent salt response of EsABI5. Although authors provided Heat map in Figure 10, that is expression profile for ABA and MAPK signaling under salt stress. If authors suggest EsABI5 responses in the ABA dependent regulation, additional expression profiles are required with/without ABA treatment under salt stress.
- In Figure 7, EsABI5 expression was highly detected in root. However, the relative expression level of EsABI5 in root under NaCl is approximately 1.5 times higher than control. That of EsABI5 expression in leaves under NaCl is approximately 3 times higher than control. Therefore, I cannot agree that EsABI5 is involved under salt stress in root.
[Minor comments]
- Figure 8 did not complete. The left panel must be non-treatment phenotypes. Authors have to add information for each panel. In the other hands, authors have to change YWT and YOT to WT and OT in left panel.
Author Response
June 14, 2021
Dear Reviewer:
Thank you very much for your comment on my paper and your request for modification.
I have already revised on my manuscript.
[Major comments]
- Even authors provided 10 proteins that interacted with EsABI5, these are selected by bioinformatics analysis. I am not sure that all 10 proteins actually interact with EsABI5. Authors need to validate the protein interactions with transcriptional activation in yeast cell directly or expression profiles by RT-PCR indirectly.
Bioinformatics analysis of the interaction relationship indicated that EsABI5 may have interactions with these 10 proteins. Except for 2 unknown proteins, the remaining 8 proteins were all related to ABA in various stress reactions. It can hypothesize that EsABI5 perhaps plays a role in stress response and is closely related to ABA. The results of bioinformatics analysis are certain reliable, but the exact results still need experimental verification. In this manuscript, the aim of bioinformatics analysis was intended to show that EsABI5 is associated with ABA and adversity. Of course, I tried my best to select genes common to ABA, MAPK pathways and 10 functional proteins for the following qPCR of wild tobacco and trangetic tobacco plants overexpressing EsABI5 gene, but some interaction proteins could not be found in the Ncbi of tobacco, or they can’t be amplified by PCR. Therefore, only SAPK3 was verified, and the results showed that the expression of EsABI5 affected the expression of SAPK3.
- I think there is no evidence for ABA-dependent salt response of EsABI5. Although authors provided Heat map in Figure 10, that is expression profile for ABA and MAPK signaling under salt stress. If authors suggest EsABI5responses in the ABA dependent regulation, additional expression profiles are required with/without ABA treatment under salt stress.
A preliminary experiment was conducted before this study, and we found that with the increase of salt stress concentration, the endogenous ABA content of Siberian wildrye gradually increased, and ABI5 is a key transcription factor in ABA signal transduction pathway. We think there are evidence for ABA-dependent salt response of EsABI5. Therefore, we didn’t study the expression of EsABI5 under the joint stress of NaCl and ABA.
- In Figure 7, EsABI5 expression was highly detected in root. However, the relative expression level of EsABI5 in root under NaCl is approximately 1.5 times higher than control. That of EsABI5 expression in leaves under NaCl is approximately 3 times higher than control. Therefore, I cannot agree that EsABI5is involved under salt stress in root.
Because we obtained the result showed EsABI5 played a negative regulatory role, in other words, the larger the increase in EsABI5, the less salt-tolerant. the relative expression level of EsABI5 in root under NaCl is approximately 1.5 times higher than control. That of EsABI5 expression in leaves under NaCl is approximately 3 times higher than control. These showed the expression level of EsABI5 gene in roots was lower than that in leaves, suggesting that EsABI5 in roots may play a role in salt tolerance. The relative expression level of EsABI5 in root under ABA is approximately 41 times lower than control. That of EsABI5 expression in leaves under ABA is approximately 7 times higher than control. And ABA content increased in response to salt stress. Therefore, we think the roots may be the key antisalt organ of Siberian wildrye.
[Minor comments]
- Figure 8 did not complete. The left panel must be non-treatment phenotypes. Authors have to add information for each panel. In the other hands, authors have to change YWT and YOT to WT and OT in left panel.
I have modified them, which are as follows:
I have modified all problems according to the requirements. If the modifications do not meet your requirements, or there are other modifications, I will be very willing to correct them.
Thank you for your consideration. I look forward to hearing from you.
Sincerely,
Fengling Shi
College of Grandland, Resources and Environment
Inner Mongolia Agricultural University, Hohhot 010011, China
E-mail: [email protected]

Reviewer 4 Report
Siberian Wildrye (Elymus sibiricus L.) Abscisic Acid-Insensitive 5 Gene is Involved in Abscisic Acid-Dependent Salt Response
Abstract: Please state what is the state of transgenesis- RNAi or over-expression and for what gene and indeed what plant? Will make it clearer for the reader when reading the abstract.
Introduction: More background information on ABA and its origins and roles is needed. More explanation of what PYR/PYL/RCAR ABA receptor is please!
Should check out: Abscisic acid - enemy or saviour in cereals response to abiotic and biotic stresses Marta Gietler, Justyna Fidler, Mateusz Labudda and Małgorzata Nykiel; Int J Mol Sci 2020 Jun 29;21(13):4607.doi: 10.3390/ijms21134607.
Results: Again, state in text what the transgenic plants are- i.e. transgenic tobacco plant over- expressing the EsABI5 where shown to be ….
What is the ploidy level of Elymus sibiricus? How many forms of the ABI5 were detected and if so, which form was the most highly expressed? This data will need to be added.
Elymus sibiricus (4x)-StH
Section 2.3- I assume you looked at ABI5 expression in the roots and leaves of Elymus sibiricus? Please state this in text and amend legends accordingly.
How many transgenic lines of tobacco did you look at- you need to look at and characterise the response of at least 3-5 independent transgenic tobacco lines- 2 is not enough!
Any vegetative data- leaf and root weights?- photographs of stressed leaves? Differences between transgenic and controls??
Materials and Methods: RNA was extracted from what? What age was the tissue used? Were they single plant and root extractions or did you pool leaf and root samples?
What accession did you use for Elymus sibiricus?; where did you obtain the seeds? Did you look at just one accession?
Were the transgenic lines you used homozygous? What method did you used for your tobacco transformation? How many weeks/months in tissue culture- what selection markers did you use for selection of putative transgenics in tissue culture.
Discussion:
A lot of the discussion could go into the introduction- it is too factually heavy- you need to discuss the results of your work in the context of previous or similar work/research being currently or historically carried out.
Author Response
June 14, 2021
Dear Reviewer:
Thank you very much for your comment on my paper and your request for modification.
I have already revised on my manuscript.
Abstract: Please state what is the state of transgenesis- RNAi or over-expression and for what gene and indeed what plant? Will make it clearer for the reader when reading the abstract.
I have modified it, which is as follows:
tobacco plants overexpressing the EsABI5
Introduction: More background information on ABA and its origins and roles is needed. More explanation of what PYR/PYL/RCAR ABA receptor is please!
I have added them in Section 2.4 , which are as follows:
Under salt stress, ABA content of cells increases rapidly, then ABA binds to PYR/PYL/RCAR receptors and triggers a second messenger Ca2+ signaling system. These bind to PP2C to form ABA-PYR/PYL-PP2C complex, releases SnRK2 protein kinase, which is inhibited by PP2C.
Should check out: Abscisic acid - enemy or saviour in cereals response to abiotic and biotic stresses Marta Gietler, Justyna Fidler, Mateusz Labudda and Małgorzata Nykiel; Int J Mol Sci 2020 Jun 29;21(13):4607.doi: 10.3390/ijms21134607.
I have added them in Section discuss, which are as follows:
The results of this study also showed that with the increase of ABA concentration, the expression of leaves EsABI5 goes up and then down, and were opposite for roots EsABI5, indicating that ABA played a positive regulatory role within a certain concentration range, and it may play a negative regulatory role beyond this concentration. This result was similary to Marta’s [70].
Results: Again, state in text what the transgenic plants are- i.e. transgenic tobacco plant over- expressing the EsABI5 where shown to be ….
I have added them, which are as follows:
2.3. Decreased Salt Tolerance of EsABI5 Transgenic Tobacco
Tobacco plants overexpressing the EsABI5 (OT) became more yellow and wilted compared with WT plants under salt stress (Figure 8).
What is the ploidy level of Elymus sibiricus? How many forms of the ABI5 were detected and if so, which form was the most highly expressed? This data will need to be added.
Although some studies have shown that there are tetraploid and hexaploid types of Elymus sibiricus, I have studied the ploidy level of E. sibiricus and E. nutans. The results show that the ploidy of Elymus sibiricus is 2n=4x=28, while the ploidy of E. nutans was 2n=6x=42.
Therefore, the ploidy of E. sibiricus in this manuscript is tetraploid and the chromosome number is 28. Two different transcription sequences of ABI5 gene were obtained during homologous cloning. NCBI BLAST showed that the similarity of the two transcripts was 96.39%, but the occurrence frequency of one of the transcripts was low. Therefore, in the later cloning of RACE, the transcripts with higher occurrence frequency were selected for 3 '- and 5' - cloning.So we had only one EsABI5 sequence.
Elymus sibiricus (4x)-StH
The genome of Elymus sibiricus is StStHH.
Section 2.3- I assume you looked at ABI5 expression in the roots and leaves of Elymus sibiricus? Please state this in text and amend legends accordingly.
I have modified them, which were all on revised manuscript.
How many transgenic lines of tobacco did you look at- you need to look at and characterise the response of at least 3-5 independent transgenic tobacco lines- 2 is not enough!
I have modified them, which are as follows:
12 transgenic lines were obtained by screening resistant culture medium and by PCR detection using specific primers. Transgenic and WT tobacco seedlings with consistent growth were selected and transplanted into flowerpots (sandy soil:nutrientsoil:vermiculite = 1:1:0.5) at 25°C with 60% humidity, 14-h/10-h light/dark cycle, and 61.64 μM·m-2·s-1 light intensity. After normal watering and culture for 2 weeks, 4 transgenic lines were detected by PT-qPCR (Primers were EsABI5-F1 and EsABI5-R1).
Any vegetative data- leaf and root weights?- photographs of stressed leaves? Differences between transgenic and controls??
I have modified them, which were all on revised manuscript.
Materials and Methods: RNA was extracted from what? What age was the tissue used? Were they single plant and root extractions or did you pool leaf and root samples?
RNA was extracted from the leaves and roots of Siberian wildrye. When the second true leaf emerged, that's the three-leaf stage. they single the leaves and roots of Siberian wildrye extraction.
I have modified them, which are as follows:
When the second true leaf emerged, some of the samples were used as controls, some were placed in 250 mM NaCl solution, and some were placed in 20, 50, or 100 µM exogenous ABA (cat.no.CA1011, Coolaber, Beijing, China) solution. Assessments of biological processes were repeated three times. When the phenotype changed, the leaves and roots of 5 different treatments were single collected, quickly frozen in liquid nitrogen, and stored at -80°C.
What accession did you use for Elymus sibiricus?; where did you obtain the seeds? Did you look at just one accession?
one accession with the strongest salt tolerance was obtained through the experiments of salt tolerance at germination and seedling stage of 20 wild Siberian wildrye.
I have modified them, which are as follows:
High-salinity Siberian wildrye (Tuzuo Banner of Hohhot City, Inner Mongolia, China, E111°27 ', N 40°47 ', 1157 m.) was cultivated to obtain seedlings for anti-salt experiments.
Were the transgenic lines you used homozygous? What method did you used for your tobacco transformation? How many weeks/months in tissue culture- what selection markers did you use for selection of putative transgenics in tissue culture.
In this study, I only want to know the mechanism of EsABI5 gene, and don't need to stabilize its inheritance, so the transgenic lines were not screened for homozygous lines.
I have added them, which are as follows:
The amplified CDS fragment of EsABI5 was successfully introduced into the vector pART-CAM, and the tobacco leaf disc was infected by the Agrobacterium-mediated method. The buds were then induced to differentiate, about 6 months of tissue culture positive lines were obtained. Polymerase chain reaction (PCR) and RT-qPCR detection of the tobacco plants overexpressing the EsABI5(OT) .
Discussion:
A lot of the discussion could go into the introduction- it is too factually heavy- you need to discuss the results of your work in the context of previous or similar work/research being currently or historically carried out.
I have modified them, which were all on revised manuscript.
I have modified all problems according to the requirements. If the modifications do not meet your requirements, or there are other modifications, I will be very willing to correct them.
Thank you for your consideration. I look forward to hearing from you.
Sincerely,
Fengling Shi
College of Grandland, Resources and Environment
Inner Mongolia Agricultural University, Hohhot 010011, China
E-mail: [email protected]

Round 2
Reviewer 3 Report
Comments to Author
Even I do not totally agree to revision version, authors’ response looks reasonable and acceptable. Therefore, I recommend this manuscript with accept after minor revision.
[Minor comments]
I think Figure 8 still does not complete. All four panels were indicated with “WT” and “OT”. Are 4 panels phenotype under salt stress? If yes, only 1 panel is required in the manuscript. I was wondering if non-treatment and salt treatment in each panel.
Reviewer 4 Report
A still feel more can be added to the introduction on the role and importance of ABA in abiotic stresses- this is still lacking and not very informative for the reader.
Author Response
June 23, 2021
Dear Reviewer:
Thank you very much for your comment on my paper again. I have already added to the introduction about the role and importance of ABA in abiotic stresses on my manuscript.
A still feel more can be added on the role and importance of ABA in abiotic stresses- this is still lacking and not very informative for the reader.
I have added them, which were all on revised manuscript.
The abscisic acid (ABA) is a vital plant hormone which orchestrates the plants in the adaptive response to abiotic stresses, such as salt, drought, and cold stresses , and regulates complicated metabolic and physiological mechanisms essential for survival in adverse environment[1-4]. ABA will quickly accumulate and cause stomatal closure to limit water loss through transpiration under abiotic stresses, as well as mobilizes a series of genes that can protect the cells from the ensuing oxidative damage in prolonged stress, and the sinaling network mediating these various responses against abiotic stresses is highly complex[5,6]. ABA is involved in complex signaling networks, including the PYR/PYL/RCAR ABA receptor, PP2C protein phosphatase, SnRK2 protein kinase, and ABI5/AREB/ABF transcription factor networks [7, 8].
Thank you for your consideration. I look forward to hearing from you.
Sincerely,
Fengling Shi
College of Grandland, Resources and Environment
Inner Mongolia Agricultural University, Hohhot 010011, China
E-mail: [email protected]